# Multi-Dimensional Evaluation Framework for the Sustainable Development of Forest Health Bases and Site Selection for Application in China

**Chenjing Fan** [1] , **Lingling Zhou** [1] , **Zhenyu Gai** [1] , **Shiguang Shen** [1,*] , **Chu Liu** [2] **and Shiqi Li** [1]

1 College of Landscape Architecture, Nanjing Forestry University, Nanjing 210037, China; fancj@njfu.edu.cn (C.F.); zllzll@njfu.edu.cn (L.Z.); gzy1997@njfu.edu.cn (Z.G.); lsq@njfu.edu.cn (S.L.)
2 Department of Construction and Real Estate, School of Civil Engineering, Southeast University, Nanjing 211189, China; 220211393@seu.edu.cn
* Correspondence: shensg@njfu.edu.cn

**Abstract:** In the context of global aging, people's awareness of health is deepening, and the rapid economic development has drawn widespread attention to the health tourism industry. As a way of experiencing health, forest health tourism is becoming increasingly favored, and the site selection and construction of forest health bases (FHBs) have also developed accordingly. To ensure sustainability in the process of the site selection and construction of FHBs, the suitability of regional development and the relative coordination of the market, environment, and resource levels should be considered. Although there have been numerous studies on sustainable forestry management, a comprehensive sustainability assessment framework based on development suitability and coordination in three dimensions is needed to guide the site selection and the construction of FHBs. The following tasks were carried out in this study: (1) based on market sustainability goals, environmental optimization goals, and ecological resource sustainability goals, a comprehensive sustainability evaluation framework for development suitability indicators and coordination indicators in three dimensions was established; (2) via the use of this framework, the construction potential of FHBs in 41,636 towns in China was evaluated; the evaluation results show that the towns in Anhui, Jiangxi, Guangdong, Guangxi, Fujian, Zhejiang, Hunan, Hubei, Guizhou, and other provinces of China generally have superior conditions for the development of FHBs; (3) a multi-dimensional comprehensive analysis of FHB site selection sustainability based on development suitability and coordination was carried out for four batches of approved pilots. The comprehensive analysis results demonstrate the worsening evaluation results of the four batches. The proposed framework can provide a reference for FHB development policies for countries worldwide.

**Keywords:** forest health market; forest health environment; forest resources; health potential; sustainability goals; coordination analysis; development suitability evaluation

## 1. Introduction

Good health and well-being have always been goals pursued by human society, and the construction of healthy public infrastructure is essential to their promotion. Increasing urbanization, irregular lifestyles, complex social relationships, insecure employment environments, and unprecedented conditions like the COVID-19 pandemic have introduced a variety of psychological and physical problems to residents [1], such as cardiovascular, metabolic, immunological, oncological, and psychiatric diseases [2]. Furthermore, the proportion of the global elderly population has continued to increase, and nearly 100 countries have entered the ranks of an aging society. Against this background, the health industry, represented by health tourism and elderly services, has gained tremendous opportunities for development [3]. According to statistics revealed by the Global Wellness Institute (GWI), the total output of the tourism industry accounted for 11% of the global GDP in

2020, while that of the health industry accounted for 12% [4]. The wellness concept is transforming almost every aspect of travel, and wellness tourism will only grow faster in the years ahead, as it lies at the powerful intersection of two massive, booming industries, namely the $2.6 trillion tourism industry and the $4.2 trillion wellness market [4]. By 2022, the scale of the health tourism market will reach 919 billion USD, with a compound annual growth rate of 7.5%, which is more than twice the growth rate of the overall tourism industry [4]. International health tourists spend an average of 1528 USD per trip, which is 53% more than typical international tourists [5]. Taking suitable measures to build healthy public infrastructure is essential to the promotion of public health and the improvement of people's well-being. Furthermore, the sustainable development of the industry should also drive the economic development of forest areas to maintain their normal operation.

Studies have identified beneficial effects of contact with natural ecosystems on disease prevention and rehabilitation [6–13]. Forests and their environments have excellent healing effects on human physiology and psychology; forest health tourism has the effects of relieving pressure, eliminating fatigue, improving the activity of immune cells, and increasing the number of anti-cancer proteins, among others [14]. A forest health base (FHB), also called a forest recreation site or forest wellness base, is an economic unit that uses the forest environment, which has healthcare functions, for the development of characteristic healthcare products and the creation of environmental space venues, supporting facilities, and corresponding service systems for tourism, food and medicine, fitness, healthcare, convalescence, cognition, experiences, and other services [15,16]. To undertake forest wellness tourism, places or facilities that can perform forest healing are constructed in the forests of different countries, and are collectively referred to as "FHBs" in the present study. Over the last few decades, relying on their abundant forests and other natural resources, some developed countries have constructed various types of FHBs in different forms and to varying degrees [17]. As the birthplace of FHBs, there are more than 350 FHBs in Germany that receive about 300,000 people each year, and the average stay per person is over three weeks [18]. In the USA, after the advancement of forest healing projects, the total national medical expenses will be reduced by 30%; one-eighth of the per capita income is used to develop the forest health industry, which receives approximately 2 billion tourists annually [19]. Japan has 63 certified FHBs covering almost all counties, and receives millions of tourists annually [20]. South Korea has established more than 300 FHBs, and one-fifth of the population participates in forest healing activities every year [21]. It is evident that, as a prosperous facet of health tourism, forest health tourism is a new direction for the sustainable development of the global forestry industry [4].

Site selection is directly related to the future sustainable development of FHBs. The experience of developed countries shows that FHB site selection that considers the health market and traffic accessibility can improve the income of forest residents, promote local economic development, and ensure the sustainable development of forest conservation [22]. A good environment is beneficial to the exertion of healing, and encourages users to coordinate physical and mental health development. The coordination of the three dimensions of society, the environment, and resources is also the basis of sustainable development, and a reasonable FHB distribution can alleviate the human–land conflict and protect ecosystem stability [23]. Therefore, the site selection evaluation of FHBs is the basis of the development and construction of the forest health industry, and whether the site selection is scientific is directly related to whether an FHB can be developed sustainably.

Some developing countries, especially those with an unbalanced distribution of forest land, rapid urbanization, and an aging population, have not yet formed a scientific and sustainable site selection evaluation system and construction guidance model. These countries and areas are prone to irrational spatial distributions and ecological damage due to the short-term blind expansion of the number of bases under construction [24]. Consider China as an example; the State and Forestry Industry Association has only issued four industry or group standards [15,16,25,26], which are only related to the nature of the entry threshold; the purpose is to review the construction of FHB projects, rather than to

ensure their sustainable development. Moreover, the forest health industry is currently in the exploration phase in terms of theoretical research, industrial practice, and regulatory guidance [27–29]. These factors may all affect the sustainable development of China's forest health industry.

Studies on the site selection and construction of FHBs have been largely focused on the evaluation of natural social conditions and management services, and have seldom considered the sustainability of the market and the human–land development coordination. This lack of consideration will cause FHB construction projects to be prone to unsustainable dilemmas, such as damage to the environment, the unsuitability of the environment, and difficulty in profitability. Thus, there is a need to establish a global, operable, and sustainable evaluation framework to support the sustainable development of FHBs. This article puts forward a new decision support framework that integrates the goals of three dimensions, namely a sustainable health market, a sustainable health environment, and sustainable forest resources, based on which a comprehensive development suitability score is calculated. Then, a coordination degree model is used to characterize the coordination degree of the three dimensions [30], and the GE matrix (McKinsey Matrix) method is finally used to evaluate the sustainability of FHBs to determine the optimal solution for sustainable development. The ultimate goal is to promote the development of forest eco-tourism and mitigate the contradiction between protection and development. Furthermore, after its establishment, the operability of the multi-dimensional framework is verified by taking China as an example; the framework is used to evaluate the development sustainability of 41,636 towns in China (based on China's town-level administrative divisions in 2019). The construction sustainability of batches I–IV of pilot FHBs that have been approved for construction is then evaluated, and the towns that have the potential to build FHBs, but have not yet done so, are ultimately identified.

## 2. Multi-Dimensional Evaluation Framework for the Sustainable Development of Forest Health Bases (MSFHB)

### 2.1. Multi-Dimensional Goals of the Sustainable Development of FHBs

The sustainable development of tourism has been extensively discussed in the past [21], and scholars have studied community tourism [31], wetland tourism [32], marine ecosystems [33], and several other factors to evaluate site sustainability. Sustainable forest management planning has also been widely discussed. For example, based on multi-dimensional decision analysis theory, Uhde et al. established an evaluation framework for forest management planning that includes ecological, social, and economic dimensions [34]. Forest health tourism is a type of recreational tourism based on the natural environment, and research on the decision-making of sustainable locations should consider not only the sustainability of the market and resource dimensions, but also the livability of the natural environment [35–37]. In addition to the development suitability of tourism, the coordination of the previously mentioned market, environment, and resource dimensions must be considered to overcome the existing shortcomings [38]. Only if forest resources for the construction of the forest health industry are developed based on a conceptual framework of sustainable development can multiple benefits be introduced to forest areas, which would be conducive to the long-term development of the forest health tourism industry [39].

According to previous research on land evaluation [40], sustainable ecological development [41], forest tourism [42], and sustainable development coordination analysis theory [43–45], a three-dimensional evaluation framework for the optimal site selection of FHBs is constructed to achieve the goals of sustainable economic development, an optimal health environment, and the ecological sustainability of forest resources. In this work, descriptive statistical methods are used to construct a frequency distribution diagram (Figure 1). The site selection framework for FHBs is divided into three evaluation dimensions, namely the sustainability of the health market (HM), the sustainability of the health environment (HE), and the sustainability of forest resources (FR), to evaluate whether the impact of an FHB on the ecological environment can be minimized and the potential

economic development can be maximized [46]. The consumption level of tourists and the infrastructure investment of local governments are important factors that affect the sustainable development of FHBs. The environmental quality of FHBs determines the healthcare effect, and will also affect the sustainable development capacity of forest tourism. FR are the basis of the development of forest tourism, and their ecological carrying capacity and resource quality are the key factors that determine the sustainable development of forest tourism. The comprehensive evaluation results of the three dimensions represent development suitability, and the difference between the development levels of these dimensions should be as small as possible to achieve a higher state of sustainable development.

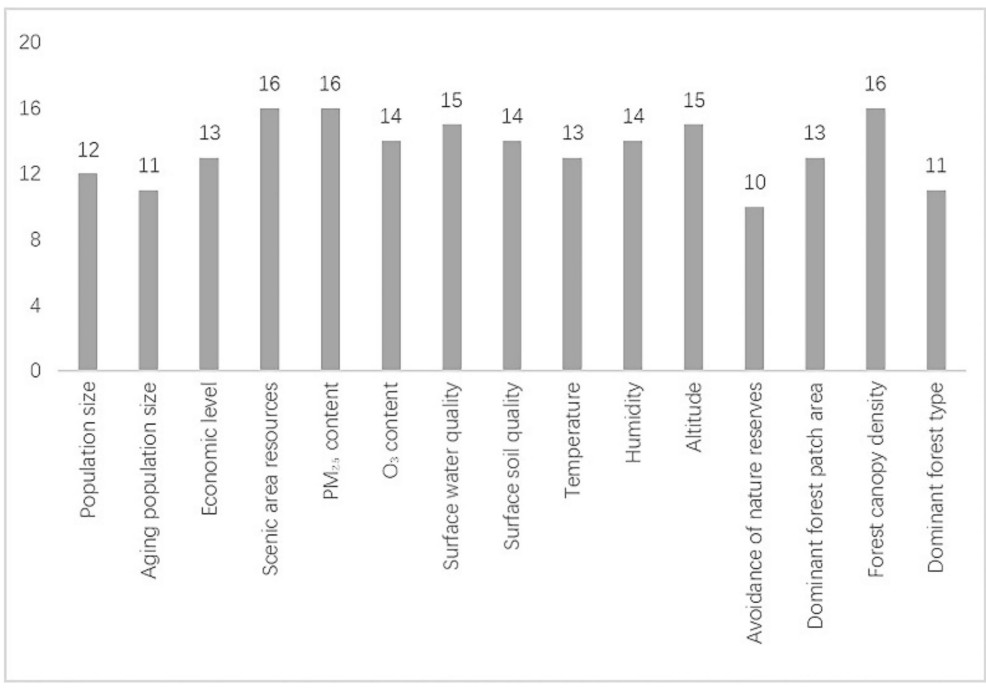

**Figure 1.** The frequency distribution diagram of FHB evaluation standards according to existing studies.

### 2.2. Framework Construction

Based on the goals of sustainable site selection and the construction of FHBs, as well as relevant regional site selection evaluation policies or norms, a comprehensive set of criteria for the operationalization of the three dimensions of sustainable goals was defined in Section 2.1. The framework selection was based on the following principles: the criteria must be representative, global, measurable, coherent, independent, cohesive, and widely available in the entire research field [39]. According to the combing of the relevant norms presented in Table 1, a list of multi-dimensional sustainable evaluation criteria was established. Each indicator is described in Table 2.

**Table 1.** A summary of the research standards for the site selection of FHBs.

| Dimension. | Indicator | Case Study | | | | | | | | |
|---|---|---|---|---|---|---|---|---|---|---|
| | | **Japan** | **South Korea** | **Russia** | **Germany** | **Australia** | **United States** | **Chinese Forestry Administration** | **Guizhou Province, China** | **Sichuan Province, China** |
| | Research object | Certification standard for forest bathing base [47] | Recuperative forest law [48] | Forest park [49] | Climate health base [50] | Recuperative forest [51] | Wellness forest [52] | National forest health base [15,16] | Code for the construction of a forest health base in Guizhou Province [25] | Standards for the evaluation of forest health bases in Sichuan Province [26] |
| Healing market | Population or aging population | ★ | ★ | — | ★ | ★ | — | ★ | — | — |
| | Transportation and markets | ▲ | ▲ | — | ★ | — | — | ▲ | ★ | ▲ |
| | Attractions | ★ | ★ | — | ▲ | ▲ | — | ▲ | ★ | — |
| Healing environment | Air quality | ▲ | — | — | ★ | - | — | ★ | ★ | ★ |
| | Water quality | — | ▲ | — | ▲ | ▲ | ▲ | ★ | ★ | ★ |
| | Temperature | ▲ | — | — | ★ | — | ▲ | ★ | — | — |
| | Humidity | ▲ | — | — | ▲ | — | — | ★ | ★ | ★ |
| | Altitude | — | ▲ | — | ★ | — | — | ★ | ★ | ★ |
| Forest resources | Avoidance of nature reserves | ★ | ★ | ★ | ▲ | ▲ | ▲ | ▲ | — | — |
| | Forest patch | — | ★ | ▲ | ▲ | ▲ | — | ★ | ★ | ★ |
| | Forest area Average canopy density of forest | — | ▲ | ▲ | ▲ | — | — | ★ | ★ | ★ |
| | Maximum patch area Dominant forest type | — | ▲ | ▲ | — | ▲ | — | ▲ | — | ★ |

Note: "—" indicates that the evaluation standard does not involve this factor; "▲" indicates that the evaluation standard involves this factor, but there is no detailed quantitative index, or there is a relative evaluation index; "★" indicates that there is a detailed quantitative evaluation standard for this dimension.

**Table 2.** The descriptions of the evaluation criteria for the sustainable site selection and construction of FHBs.

| Factor | Indicator | Description |
|---|---|---|
| Population size ($X_{HM1}$) | Population within a certain range of travel time [27] | Health markets with a large population within the scope of the source market are large |
| Aging population ($X_{HM2}$) | Elderly population within a certain range of travel time [16] | Areas with a large number of elderly people in the tourist market have a broader health market |
| Economic development level ($X_{HM3}$) | Per capita GDP within a certain range of travel time [53,54] | Areas with high per capita GDP have a wider market for tourism consumption, but markets in low-development areas are not saturated and also have potential |
| Scenic area resources ($X_{HM4}$) | Scenic area resources [51] | Administrative scope of a town with a pleasant view is more attractive to people |
| PM$_{2.5}$ ($X_{HE1}$) | The concentration of PM$_{2.5}$ [25] | Places with low PM$_{2.5}$ concentrations have better air quality and are more livable (forest area) |
| O$_3$ ($X_{HE2}$) | The concentration of O$_3$ [25] | Places with good air quality are more livable; the threshold value is when the concentration of near-ground O$_3$ reaches 100 µg/m$^3$ (forest area) |
| Surface water quality ($X_{HE3}$) | Surface water quality [26] | The surface water quality reaches the national or international standard II or above (forest area) |
| Temperature ($X_{HE4}$) | Temperature; number of 15–25 °C days in a year [15,16] | Temperatures between 15–25 °C are considered comfortable, and a longer comfortable period is more conducive to forest health tourism activities (forest area) |
| Humidity ($X_{HE5}$) | Relative humidity; number of 45% to 65% RH days in a year [15,16] | Air is more comfortable at a RH of 45%–65%, and a longer comfortable period is more conducive to the operation of forest health tourism activities (forest area) |
| Altitude ($X_{HE6}$) | Altitude [50] | Altitudes below 1500 m are good for human health, whereas excessively high altitudes can easily cause altitude sickness (forest area) |
| Avoidance of nature reserves ($X_{FR1}$) | Nature reserve [51] | Avoiding national natural protection areas is beneficial to the sustainable development of forest resources |
| Dominant forest patch area ($X_{FR2}$) | Contiguous area of forest patches [48] | The base should have a centralized contiguous forest area |
| Forest canopy density ($X_{FR3}$) | Average canopy density of the maximum forest patch area [24] | The concentrated and contiguous forest coverage rate at the base should be high |
| Dominant forest type ($X_{FR4}$) | The dominant forest type is a natural forest [55] | Natural forests with a complete forest structure and tree species structure are preferred as FHBs |

*2.3. Evaluation Method*

It is necessary to conduct further quantitative scoring of the site selection sustainability of FHBs to screen out the areas with the most sustainable development space. To quantitatively measure the development sustainability of FHBs, the development suitability and coordination of the HM, HE, and FR dimensions must be comprehensively considered via the following three steps: (1) development suitability evaluation; (2) coordination evaluation; (3) comprehensive sustainability evaluation.

### 2.3.1. Development Suitability Evaluation

The suitability of each dimension is evaluated based on the principal component analysis method [42], after which the comprehensive development suitability is calculated. First, the raw data of the indicator layer must be non-dimensionalized (standardized) using Equation (1) or (2), such that the positive and negative indicators are comparable:

$$U_{(x)} = \frac{(X - X_{min})}{(X_{max} - X_{min})} \text{ Positive indicators } (+), \tag{1}$$

$$U_{(x)} = \frac{(X_{max} - X)}{(X_{max} - X_{min})} \text{ Negative indicators } (-), \tag{2}$$

Where $U_{(x)}$ is the standardized indicator value of the HM, HE, or FR dimension, $X$ is the measured indicator value, and $X_{max}$ and $X_{min}$ are the maximum and minimum raw values of an indicator, respectively. The standardized indicator value is between 0 and 1. Then, dimensionality reduction analysis is performed on the standardized evaluation values of each dimension to extract the principal components of the evaluation factors of each dimension. According to the variance contribution rate, the number of principal components is determined, and the linear weighting method is used to calculate the single-dimensional comprehensive principal component evaluation values $F_{HM}$, $F_{HE}$, and $F_{FR}$. $\overline{F}$ is the arithmetic mean of the principal component evaluation values of the three dimensions and can be calculated as the evaluation result of the development suitability of the FHB.

### 2.3.2. Coordination Evaluation

The HM, HE, and FR dimensions are multi-level and open systems. There are complex relationships between these dimensions, and their coordination without development shortcomings is the premise of sustainable development. To quantify the coordination degree of these dimensions, the coupling coordination degree model (CCDM) used in physics is adopted [38]. The calculation formula is given by Equation (3):

$$S_i = \sqrt{\frac{1}{3}\left[\left(F_{HM} - \overline{F}\right)^2 + \left(F_{HE} - \overline{F}\right)^2 + \left(F_{FR} - \overline{F}\right)^2\right]} \tag{3}$$

$$C_i = 1 - S_i \div \overline{F} \tag{4}$$

where $F_{HM}$, $F_{HE}$, and $F_{FR}$ are the single-dimension comprehensive principal component evaluation values, the arithmetic mean of which is $\overline{F}$. Moreover, $S_i$ is the standard deviation of the performance values of the development index for the HM, HE, and FR dimensions, and $C_i$ is the coordination evaluation values of the three dimensions.

### 2.3.3. Sustainability Evaluation

In this work, the FHB sustainability evaluation matrix is constructed by combining the GE matrix of the development suitability index (DI) and the coordination index (CI) proposed by Shen et al. [38]. Based on this matrix, the local government can judge whether there is a sustainable development problem in the construction of a local FHB. Such problems can be divided into two categories. In the first case, there is no development advantage (areas with DI values in the GE matrix less than 0.5). In the second case, there are development shortcomings (areas with CI values in the GE matrix less than 0.5). As

shown in Figure 2, regions i, ii, and iv are characterized by a low development level and low coordination. Most of these areas are dominated by farmland and wasteland, and the construction of FHBs should not be considered. Some areas have sustainable development shortcomings at both the development and coordination levels, such as areas with low development suitability but a high degree of coordination (region iii), and areas with high development suitability but a low degree of coordination (region vii). Areas with moderate development suitability and coordination (region v) may have shortcomings in the dimensions of the environment, resources, or economy, or do not have development advantages. When building an FHB, local governments should take targeted measures to compensate for the shortcomings and enhance the well-being of all dimensions to improve the development suitability and coordination of the three dimensions. Areas with high development suitability and strong coordination (regions vi, viii, and ix) are generally considered to be the best options for FHB development, and pilot units should prioritize construction in these areas.

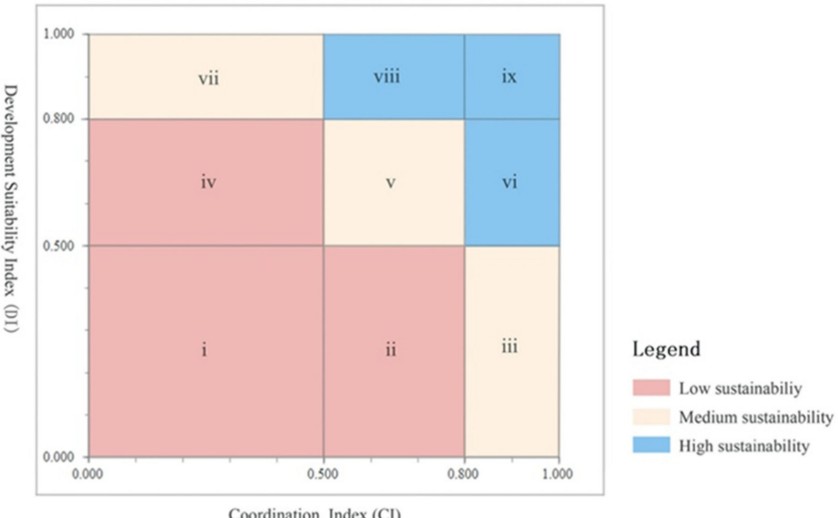

**Figure 2.** The GE matrix of the DI and CI of FHB sustainability.

### 3. Application of the MSFHB in China

*3.1. Current Situation of China's Forests and FHB Construction*

The study area considered in this research is China (Figure 3). With the rapid development of the Chinese health industry and forestry industry in recent years, there has been an increasingly urgent demand for the construction of FHBs [56]. On the one hand, China is characterized by a large population base and rapid economic development, as well as a huge health market. On the other hand, China has a vast land area and diverse climatic conditions, and its FR ranks fifth among the countries with the richest forest resources in the world; with $2.08 \times 10^8$ hectares of forest area, the forest coverage rates of Fujian, Jiangxi, Zhejiang, Guangdong, Hainan, and Sichuan provinces exceed 30%, thereby providing a rich material basis for the development of the forest health industry [57]. However, although the absolute amount of available forest is large, the per capita forest area is only 0.15 hectares. Because the protection of forest ecosystems has always been placed at a high position, China's land policy reflects the coexistence of privatization and public ownership. The vast majority of China's forest land is public, and a portion of it is privately owned. In China, the use of forest land is the same as that in most countries; whether the land is privately owned or not, it is almost always supervised by government departments. FHBs are reported by local enterprises or institutions and approved by the government after reviewing the relevant standards, and can only be developed and utilized after their construction is approved. As a result, China has been relatively cautious in its development of forestry, and the construction of FHBs has been carried out within a small scope [58]. At the national level, from 2016 to 2018, only four batches of 368 pilot construction units of

national FHBs were approved and publicized by the government according to the relevant standards [15,16,25,26]; these pilots included 36 units in batch I (September 2016), 99 units in batch II (June 2017), 98 units in batch III (November 2017), and 135 units in batch IV (October 2018) [59–62]. The selection of pilot units for Chinese FHBs is gradually increasing, as is the construction of existing pilots [63]. For the sustainable development of the forest health industry, it is necessary to take into account the limitations of forest resources and the stability of the ecosystem, and to establish clear site selection and construction standards and a certification system, to ultimately raise the entry threshold for the construction of FHBs.

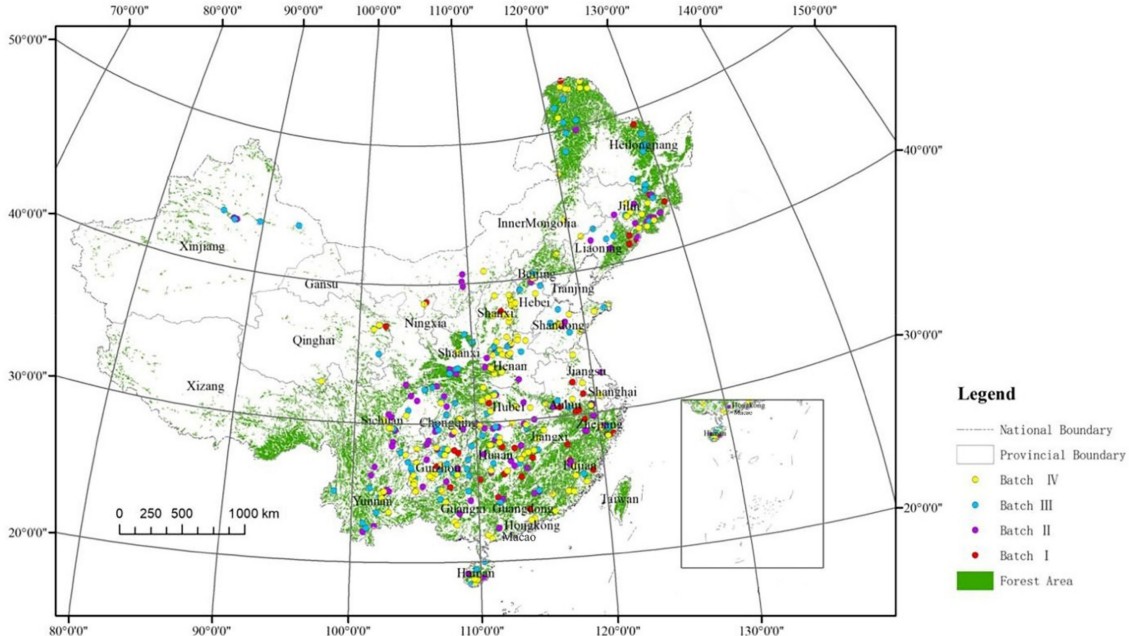

**Figure 3.** The study area.

Based on the evaluation method presented in Section 2.3, a comprehensive suitability evaluation is carried out on 41,636 Chinese towns. Moreover, the sustainability of the site selection and construction of each batch of pilot units is evaluated, and policy recommendations are put forward based on the discussion of the evaluation results. Figure 3 exhibits the map of the spatial distributions of China's forest resources and the existing pilot FHB units.

*3.2. Data Collection*

According to the sustainability evaluation framework exhibited in Table 2, the following data were collected. (1) HM dimension data: The traffic data were sourced from the Baidu Map API, the population and GDP data were sourced from the 2017 global 1-km grid precision population density dataset provided by the Land Scan platform and China's kilometer grid dataset of the spatial distribution of the GDP provided by the Resources and Environment Data Cloud Platform of the Chinese Academy of Sciences (https://www.resdc.cn, accessed on 25 June 2021). The aging population data were sourced from the seventh population census of 2020. (2) HE dimension data: Air quality data were sourced from ChinaHighPM$_{2.5}$ [63] and ChinaHighO$_3$ [64] with 1-km accuracy produced by the Atmospheric Remote Sensing Team of the Department of Atmospheric Oceanography and Science, University of Maryland. The water quality data were sourced from a report published by the China Environmental Monitoring Station (http://www.cnemc.cn, accessed on 30 March 2021), and the climate condition data were sourced from the daily monitoring data of Chinese meteorological stations in 2019 (http://data.cma.cn/en, accessed on 30 March 2021). (3) FR dimension data: Data on the forest land distribution, canopy density, and forest type were sourced from the data of the seventh forest land

survey released in 2017 in China, and data on scenic and historic interest areas and national protected areas were sourced from the public data of government departments. In addition, to verify the sustainability evaluation of the selection of China's four batches of pilot sites, data on the four batches of pilot sites were collected from the open data of the China Forestry Industry Association [59–62].

## 4. Results

### *4.1. Evaluation of Development Suitability*

4.1.1. Evaluation Results of the HM Sustainable Development Dimension

The HM development suitability was calculated according to the principal component evaluation method presented in Section 3.2, as shown in Figure 4. As is evident from Figure 4, the towns in various provinces south of the Heihe–Tengchong Line have broad prospects for economic development and high suitability for sustainable FHB development, especially in China's Beijing–Tianjin–Hebei, Yangtze River Delta, Pearl River Delta, and other regions.

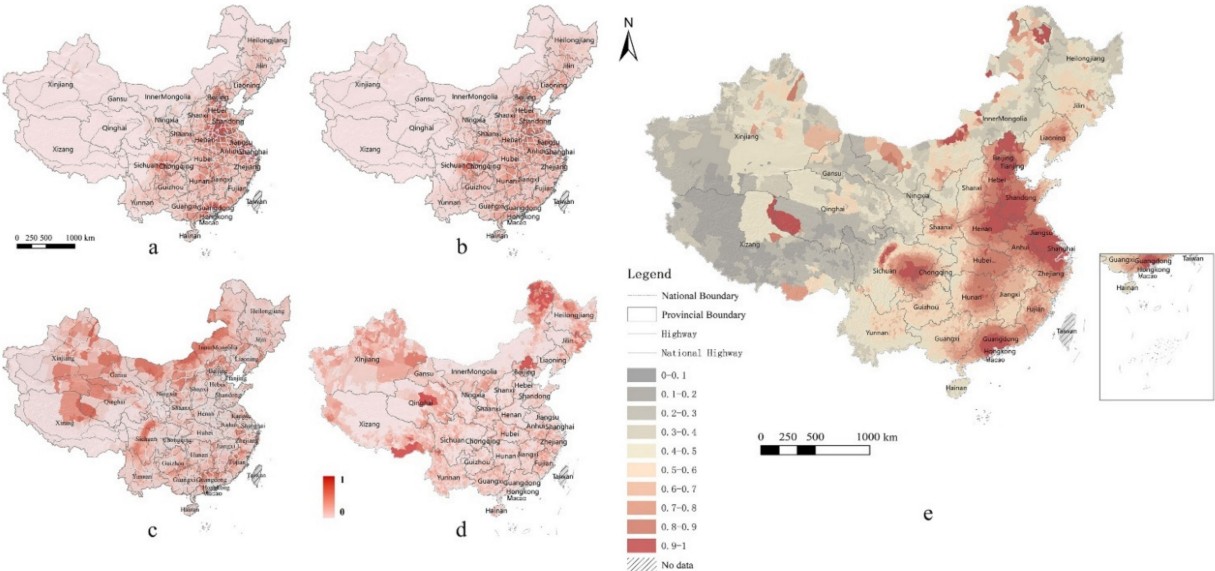

**Figure 4.** The sustainability evaluation results of the HM dimension. The evaluation results of (**a**) the total population, (**b**) the aging population, (**c**) the per capita GDP, and (**d**) scenic spots; (**e**) the DI of the HM dimension.

4.1.2. Evaluation Results of the HE Sustainable Development Dimension

The HE dimension is only used for the evaluation of the townships where forest land is located to avoid the environmental interference of the urban built-up areas. As calculated by the principal component evaluation method presented in Section 2.3, the HE dimension development suitability results are shown in Figure 5. As is evident from the comprehensive overlay results presented in Figure 5, towns in some Chinese provinces on the southeast coast such as Yunnan Guangxi, Guangdong, Hainan, Fujian, and Zhejiang were found to have the best evaluation results.

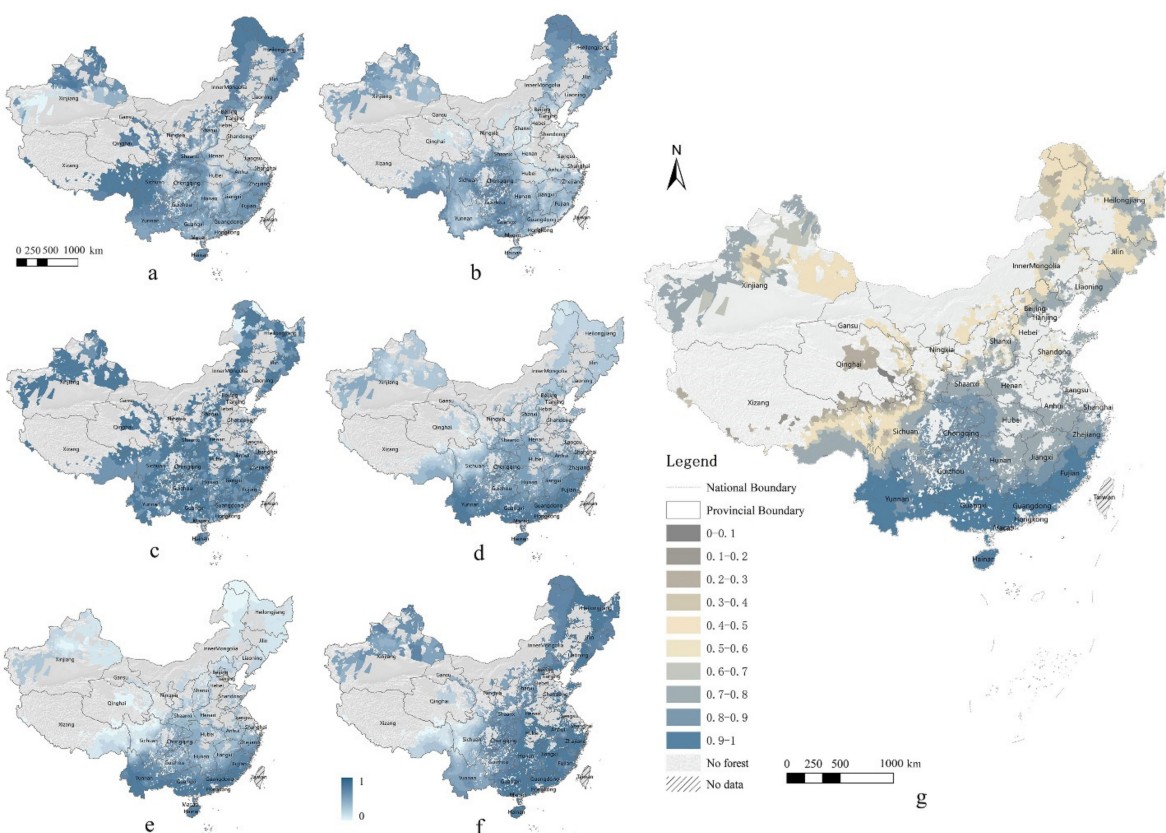

**Figure 5.** The sustainability evaluation results of the HE dimension. The evaluation results of the (**a**) water quality level, (**b**) altitude, (**c**) PM$_{2.5}$ density, (**d**) O$_3$ density, (**e**) relative humidity, and (**f**) temperature; (**g**) the DI of the HE dimension.

### 4.1.3. Evaluation Results of the FR Sustainable Development Dimension

The FR dimension development suitability was calculated according to the principal component evaluation method presented in Section 2.3, and the results are shown in Figure 6. According to the comprehensive evaluation results exhibited in Figure 6, provinces such as Zhejiang, Jiangxi, Guangdong, Guangxi, Sichuan, Yunnan, Heilongjiang, and Jilin were found to have better evaluation results in the FR dimension, and therefore have higher development suitability. In contrast, the forest resources in provinces of Northwest China are relatively scarce. Besides, some provinces such as Beijing, Tianjin, Shanxi, and Shandong have forest resources but a relatively poor forest structure and poor resource sustainability.

### 4.1.4. Comprehensive Evaluation of Development Suitability (CI)

The linear weighting method was used to calculate the comprehensive evaluation value of the development suitability (DI) of FHBs in three dimensions, and the comprehensive evaluation of the development potential of FHBs in China was carried out. As exhibited in Figure 7, areas with high suitability for FHB development were found to be concentrated in the forest villages of Guangdong, Guangxi, Fujian, Zhejiang, Guizhou, and Sichuan provinces.

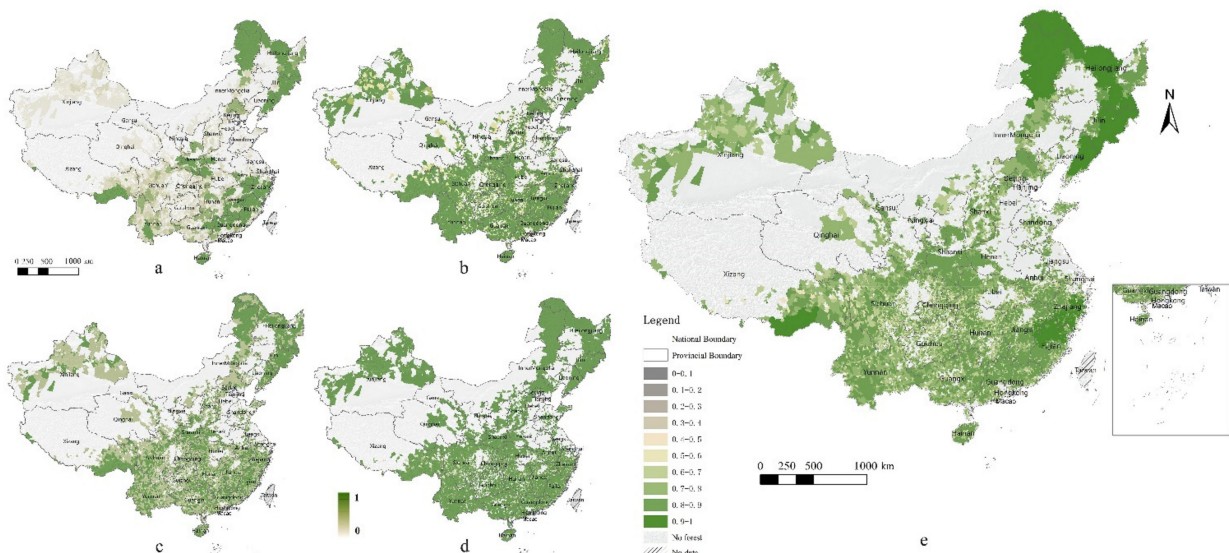

**Figure 6.** The sustainability evaluation results of the FR dimension. The evaluation results of (**a**) the forest patch area, (**b**) the forest canopy closure, (**c**) the forest type, and (**d**) whether the forest is a national protected area; (**e**) the DI of the FR dimension.

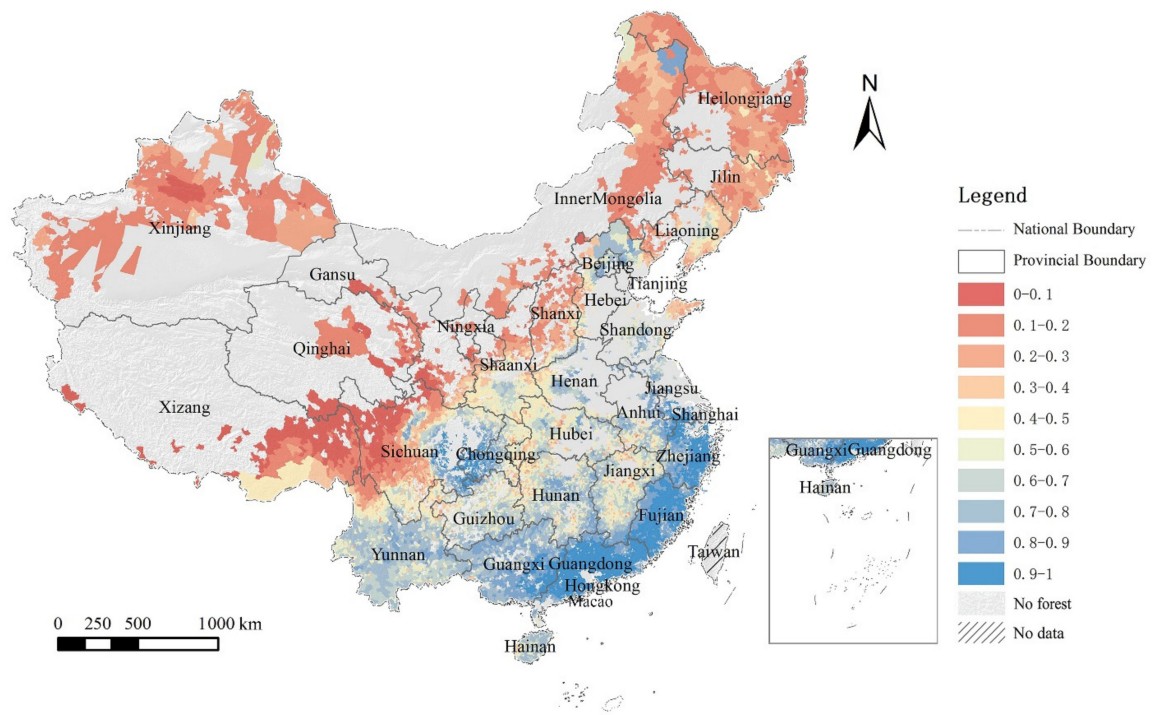

**Figure 7.** The development suitability (DI) of FHBs in China.

*4.2. Coordination Evaluation (CI)*

According to the method presented in Section 2.3 and the evaluation results reported in Section 4.1, the CCDM was used to calculate the development coordination (CI) of the three dimensions. The evaluation results ranged from 0 to 1, as shown in Figure 8. Figure 8 reveals that the development of numerous areas in Guangdong, Fujian, Zhejiang, Hunan, Hubei, and Sichuan provinces is highly coordinated, and there are no major shortcomings in the three dimensions.

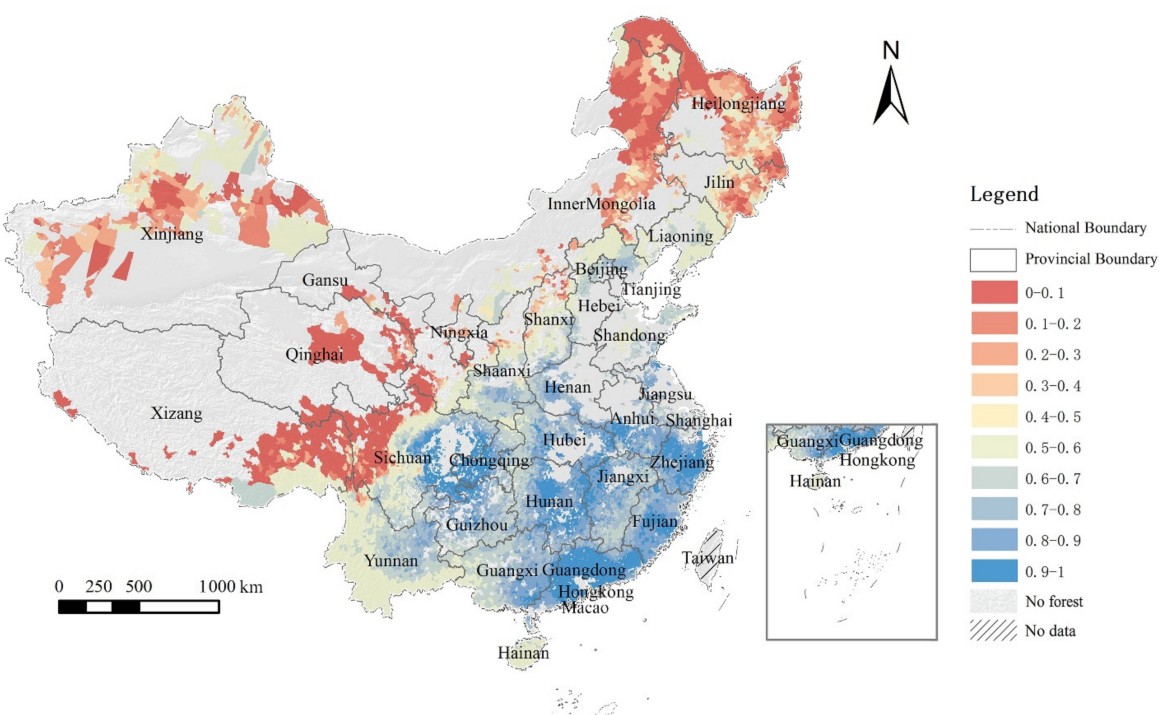

**Figure 8.** The coordination (CI) of the development of FHBs in China.

### 4.3. Sustainability Evaluation

#### 4.3.1. Sustainability Evaluation of the Construction of FHBs

In combination with the GE matrix, the comprehensive evaluation of the potential sustainability performance of the construction of FHBs in these townships with forest land was carried out. The sustainability of 41,636 townships in China was comprehensively evaluated based on the dimensions of the HM, HE, and FR (Figures 9 and 10). It can be seen from Figure 9 that 10% of the forest villages in China are characterized by high sustainability for FHB construction, and these areas are concentrated in some provinces along the southeast coast and in the middle of the country, such as Sichuan, Chongqing, Guangxi, Guangdong, Fujian and Zhejiang. Moreover, the development suitability and coordination of the three dimensions was found to be poor in 22% of the forest villages, such as Helongjiang, Jilin, Inner Mongolia, Liaoning, Xinjiang, Gansu, Ningxia, Qinghai, and Xizang. Figure 10 presents a cluster diagram of the assessment of the FHB development potential and the number of existing pilot FHBs for each province in China. As can be seen from Figure 10, the existing pilots are mostly concentrated in Guizhou, Hunan, Sichuan, Hebei, Yunnan, Jilin, Zhejiang, Heilongjiang, and other provinces, but in some provinces with better comprehensive sustainability coordination evaluation results such as Guangdong, Fujian, Jiangxi, and Guangxi province, the number of FHB pilots is not higher.

#### 4.3.2. Evaluation of the Four Batches of Selected Pilot Units

According to the evaluation methods presented in Section 2.3, Figure 11 presents the distribution matrix of the sustainability evaluation results of the development suitability and coordination of each batch of pilot units. It can be seen that most of the pilot units in these four batches are characterized by high sustainability (located in regions vi, viii, ix). It is also worth noting that a few pilot units with low sustainability, which are scattered in areas of regions i, ii, and iv, are concentrated in batches III and IV.

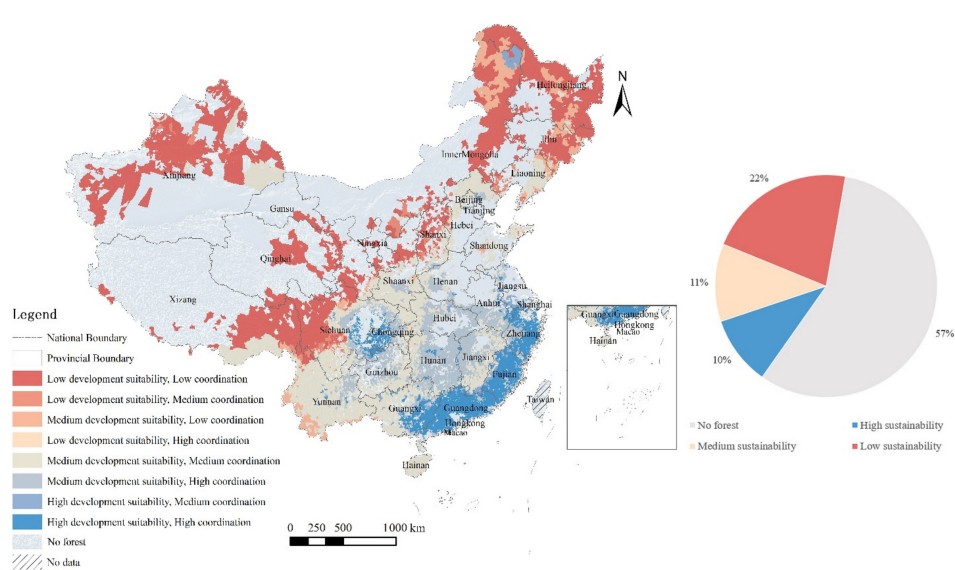

**Figure 9.** The comprehensive sustainability coordination evaluation results.

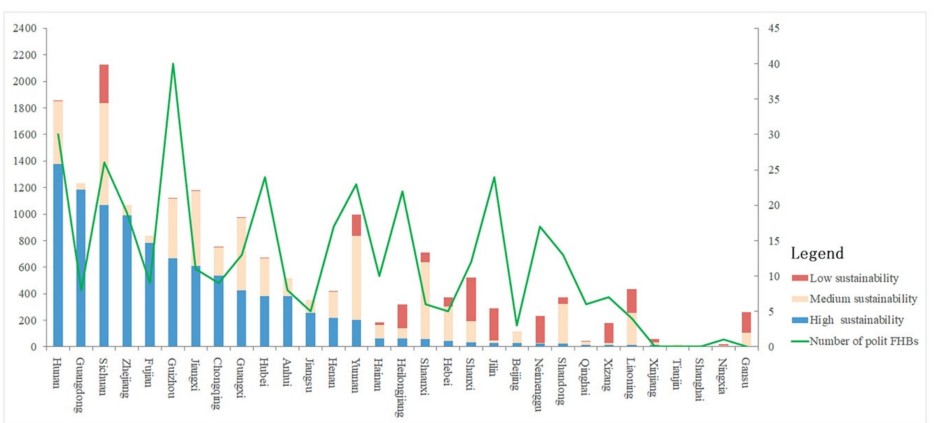

**Figure 10.** The FHB sustainability potential and numbers of the existing batches of pilot units in various provinces of China.

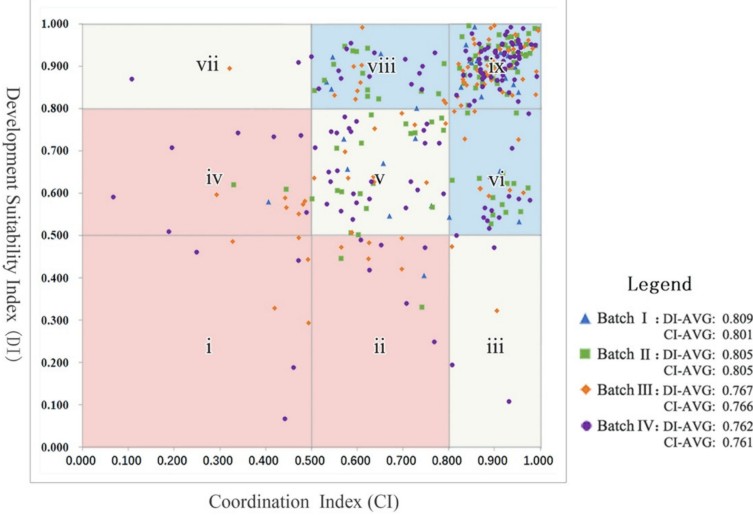

**Figure 11.** The GE matrix diagram of the development suitability and coordination of batches I–IV of pilot FHBs in China.

## 5. Discussion

### 5.1. Discussion of the MSFHB

In forest management planning, it is a challenging task to develop the forest health industry to limit the negative impacts on resources and the ecological environment while maximizing the benefits. It is necessary to balance the sustainability of the economy, the ecological environment, and resources in the selection of areas for the development of the forest health industry, and to promote its sustainable, stable, and healthy development [65]. Based on the concept of multi-dimensional decision-making [34], the sustainability goals of the HM, HE, and FR were integrated in the present work to establish the MSFHB. Then, site selection evaluation was carried out for the analysis of the development suitability and development coordination of the construction of FHBs in different dimensions to guide the site selection and layout of China's FHBs, and to promote the sustainable development of the forest health industry. The two main contributions of MSFHB include the following. (1) The framework was proposed on the basis of the HE assessment [66]. It fully reflects the understanding of limited natural resources, and takes both the market [67] and ecological sustainability [68] into account. (2) This framework adopts objective and comprehensive evaluation criteria to find areas with high development suitability, high coordination with less human–land contradiction among all dimensions.

In conclusion, compared with the traditional evaluation methods for FHBs, the proposed framework comprehensively considers the health market potential of forests, environmental livability, and the efficient utilization of resources. A GE matrix of sustainability evaluation based on development suitability and coordination was established to determine whether a certain area has development advantages and coordination ability for the construction of FHBs. The use of this method can also prevent the over-emphasis of development advantages in the process of site selection. It is beneficial to help people in areas with development shortcomings to improve their well-being via the construction of FHBs. The results visually verify the operability of this framework, rather than solely its conceptual worth. The MSFHB can also be used to effectively analyze and evaluate the potential of existing pilot FHBs and the potential for developing FHBs at the national level.

### 5.2. Discussion of the Construction of FHBs in China

(1)  Evaluation of development suitability in China

By applying the proposed MSFHB to China, the results revealed clear findings regarding the evaluation of sustainable site selection from different dimensions.

First, the three sustainability dimensions of FHBs were evaluated. Figure 4 reveals that the areas south of the Heihe–Tengchong Line in China have good forest health industry development prospects [69]. These areas have better traffic conditions and richer landscape resources, and are the best choice for the site selection of FHBs. In contrast, some of China's northwest regions, which do not have a sustainable health market, scored the worst; Figure 5 indicates that the southeast coastal areas of China have the best environmental sustainability evaluation results due to their low proportion of heavy industries, low pollution, and superior climate [45]. Moreover, some regions in northern China, such as the provinces of Heilongjiang, Jilin, Liaoning, Shandong, and Shanxi, cannot provide a good environment due to intensive heavy industry or a cold climate, and in northwest China, the comfortable period for forest health activities is shorter, which is not conducive to the sustainable development of FHBs [70]. According to the evaluation results from the perspective of FR (Figure 6), it was found that the northeastern, central, and southern regions of China have superior resource endowments for the construction of FHBs. However, the northwest region was found to have few forest resources, unstable forest ecosystems, and poor basic conditions for FHB construction.

In general, areas with high development suitability must have broad health markets, livable environments, and high FR endowments [42]. Areas with high suitability for FHB development were found to be concentrated in the forest villages of Guangdong, Guangxi, Fujian, Zhejiang, Guizhou, and Sichuan provinces (Figure 7).

(2)　Coordination evaluation in China

Whether FHBs can be sustainably developed depends not only on high suitability, but also on the high coordination of the regional society, environment, and resources [71]. Figure 8. reveals that the development of numerous areas in Guangdong, Fujian, Zhejiang, Hunan, Hubei, and Sichuan provinces is highly coordinated, and there are no major shortcomings in the three dimensions. However, in many towns in Qinghai, Xizang, Inner Mongolia, Heilongjiang, and other provinces, the coordination of the three dimensions is extremely poor. Local governments should take targeted measures to improve the level of coordination or stimulate the market by providing more infrastructure. Alternatively, they should strive to create a healthy forest environment or increase the forest area, improve the forest quality, ensure sustainable forest ecosystems, promote coordinated regional development, and enhance the potential of FHB construction.

(3)　Sustainability evaluation of FHBs in China

It can be seen from Figure 9 that 21% of forest villages in China have high sustainability for FHB construction, indicating that there are enough high-quality forest villages to develop forest health tourism, and that China's future forest health tourism development prospects are bright. However, 22% of forest villages have poor development suitability and coordination among the three dimensions (Figure 9), and it is therefore not recommended to develop the forest health industry in these areas. Regarding the medium-sustainable areas (Figure 9), due to their lack of tourism markets, poor living environments, or shortage of resources, it is recommended that they improve their development suitability and coordination before construction. It is not advisable to rush to build FHBs and follow in the footsteps of other regions, which would contribute to increased human–land contradiction. It is also worth noting that the forest villages in Guangdong, Zhejiang, Fujian, Jiangxi, and other provinces have good basic conditions for FHB development, but the number of pilot projects is unsaturated (Figure 10). In the future, the construction of FHBs can be increased in a targeted manner.

(4)　Analysis of the sustainability evaluation results and evaluation of existing pilot FHBs

The evaluation results of the existing four batches of pilot FHBs are not very impressive (Figure 11). A small number of pilots are characterized by the problems of low development suitability or poor regional coordination. Batches III and IV of pilot units have the most prominent problems. This is because, in the initial forest health industry development period, the foundation of financial support is strong and institutions are easily driven by profits to blindly invest in the forest industry [72]. The institutions tend to exaggerate their environmental conditions and resource capabilities and make blind declarations when they have an incomplete understanding of the construction of FHBs. This leads to the tendencies of the commercialization, real estate, and homogenization of recreation services, and the failure to realize economic benefits [46]. In 2017, numerous FHBs were indiscriminately selected and developed in disorder, which aggravated the loss of forest resources and was not conducive to the sustainable development of the forest health industry. In addition, the hotspots of FHBs are mainly concentrated in the coastal areas with good forest resources or a developed economy, and the uneven spatial distribution of the bases easily causes the waste of forest resources, which is not conducive to the long-term development of the forest health industry [73]. Thus, for the future site selection of FHBs, the focus should be placed on regional coordinated development and the avoidance of building FHBs in a flurry [74].

*5.3. Policy Suggestions for FHB Construction*

First, FHB construction should be developed according to local conditions [75]. Taking China as an example, regarding the provinces and cities in central China and on the southeast coast, including Hunan, Guizhou, Sichuan, Zhejiang, Fujian, Jiangxi, and Guangxi, the forest villages in these areas have the best conditions for building FHBs. There are no shortcomings in the development FHBs, and if the market is not saturated, the construction of FHBs can be vigorously developed [76]. For areas with high suitability but poor

coordination, e.g., the forest villages of Hunan and Hubei, the development shortcomings should be considered, and it is recommended that stricter access thresholds be set to avoid wasting resources and aggravating human–land conflicts driven by capital. FHB construction in areas with general suitability but high coordination, e.g., some forest villages in Yunnan and Sichuan, should be based on the principle of sustainable resources and the more sustainable regional decentralized construction of FHBs [77]. Most areas with low suitability and coordination, such as forest villages in Xizang, Gansu, Ningxia, and other provinces, are largely made up of farmland and wasteland, and the construction of FHBs is not considered. However, from the perspective of the promotion of public well-being and ensuring the coordinated development of regions, pilot studies can be conducted in these areas with the best comprehensive evaluation results to establish FHBs according to the MSFHB [78]. Additionally, at the beginning of the construction of FHBs in areas where forest resources are stable, a detailed assessment of the operational, organizational, and safety risks during timber harvesting in the area should be made with the proposed framework. For centralized governments or developing countries, the process of issuing forest health and wellness site construction indicators should be specified, and more development indicators should be made available for use in more suitable areas. In addition, considering the conflict between forest leisure tourism and forest management, it is suggested that FHBs be reported at the beginning of construction and pass inspections related to the relevant laws and regulations. It is also suggested that the proposed framework can be used to select individual towns with forest land for experimental reference. In general, considering the regional differences in the different dimensions of sustainable development, the GE matrix should be used to prioritize areas with both development advantages and higher coordination [79].

Second, a pilot evaluation system should be established by combining both top-down and bottom-up methods [80]. Taking China as an example, the forest health industry has made great progress in just four years; starting in 2016, the government has selected a stream of FHB pilot units for development and construction [59–62], and government departments are also constantly refining their criteria for the selection of FHB pilot units [15,16,25,26]. However, these criteria remain based on a selection method in which each applicant is independently inspected according to the standards set by the government and reports to higher-level government authorities. There is a lack of top-down and bottom-up coordination, as well as a lack of an efficient evaluation system and method for FHBs [81], resulting in numerous deficiencies in the operation of the constructed and operated FHBs. In the future, a stricter access system should be established to ensure that areas with better basic conditions have more opportunities for pilot projects and construction.

## 6. Conclusions

People's growing awareness of health, rapid economic development, and demographic changes have led to widespread social concern for the elderly and health service industry, and forest health tourism is becoming increasingly favored as a health experience model. Proper site selection and development based on multidimensional decision-making can enable residents to heal their bodies and minds in the best HE under the premise of sustainable ecological forest resources, and will boost the economic development of forest areas. In this article, the traditional evaluation criteria for FHBs were optimized, and a comprehensive sustainability evaluation framework (MSFHB) based on the development suitability and coordination evaluation in the HM, HE, and FR dimensions was proposed and applied to China. The development and construction potential of 41,636 towns in China were evaluated, and it was found that 21% of the towns, which are mainly distributed in Hunan, Guangdong, Sichuan, Zhejiang, Guizhou, and Jiangxi provinces, have sufficiently high sustainability to build FHBs. Furthermore, via the analysis of batches I–IV of pilot FHBs, it was found that the comprehensive evaluation results of the pilot FHBs became worse in the later batches, thereby revealing the blind and potentially capital-driven nature of the previous site selection process of pilot FHBs.

The proposed sustainability evaluation system of FHB construction provides a scientific reference and theoretical framework from the perspective of the sustainable construction capacity of FHBs. Based on the GE matrix, it can be determined whether a certain area has development advantages and coordinated capabilities for FHB construction. The use of this method can also avoid the over-emphasis of development advantages in the process of site selection, which is beneficial to improving the well-being of people in areas with disadvantaged development via the construction of FHBs. Moreover, this idea can provide a complete site selection and construction basis for countries and regions that are developing the site selection of FHBs. In this study, the shortcomings of the existing pilot bases in China were exposed, and policy support via the combination of central and local governments was provided for the selection of future pilots. However, the proposed framework is characterized by some shortcomings due to various reasons. For example, the number of evaluation factors is limited, and other influencing factors may not have been taken into consideration. Moreover, the data collection process was characterized by objective problems such as timeliness and accuracy. Additionally, the calculation of the market share, forest restoration effects, and forest ecological losses after site construction were not considered in the site selection of FHBs; these will be important components of further standard-setting and research.

**Author Contributions:** Conceptualization, C.F.; methodology, C.F. and L.Z.; software, L.Z.; validation, Z.G., S.L. and C.L.; formal analysis, C.F.; resources, L.Z. and C.L.; data curation, L.Z. and C.L.; writing—original draft preparation, L.Z.; writing—review and editing, C.F., L.Z. and Z.G.; funding acquisition, C.F. and S.S. All authors have read and agreed to the published version of the manuscript.

**Funding:** This research was funded by the National Natural Science Foundation of China (51908309, 31971721 and 31570703) and Beijing Outstanding Young Scientist Program (JJWZYJH01201910003010).

**Conflicts of Interest:** The authors declare no conflict of interest.

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
