# Peer review of "Multi-Dimensional Evaluation Framework for the Sustainable Development of Forest Health Bases and Site Selection for Application in China"

_forests, doi:10.3390/f13050799_

Round 1

Reviewer 1 Report

dear authors
Your article is very interesting. I have reservations about the presentation of the research results. In the results chapter, we have many figures and tables, a repetition of the methodology of how the given results were achieved and one sentence (in each subsection) describing the result. I think that more important results can be presented descriptively.

Author Response

Thank you for your valuable advice. With reference to your suggestion, we have deleted the redundant descriptions in the results chapter. At the same time, more important results have been presented descriptively, and we focused on clarifying some vague description results, such as some provinces, we changed to specific and clear areas.

Reviewer 2 Report

I recommend improving the quality of the figures (higher resolution is needed).

Author Response

Thank you for your valuable advice. We carefully checked and modified each figure, and the quality of the figures have been improving.
